# Plasma exchange and radiation resensitize immunotherapy-refractory melanoma: a phase I trial

Immune checkpoint inhibitors (ICI) are effective for advanced melanoma. However, most develop ICI resistance. Tumor-derived soluble PD-L1 (sPD-L1) and other immunosuppressive factors drive resistance. We hypothesized that therapeutic plasma exchange (TPE) may remove sPD-L1 from circulation and overcome ICI resistance. Patients with metastatic ICI-resistant melanoma and elevated sPD-L1 received radiotherapy to a minority of metastatic lesions, TPE, and ICI re-challenge. Primary endpoints were adverse events and sPD-L1 reduction. Secondary endpoints included overall survival, response, and progression-free survival. Correlative studies included changes in sPD-L1, other immunosuppressive factors, and immune cell phenotypes. Eighteen patients were included. Treatment was well-tolerated, and levels of sPD-L1 were reduced by TPE (mean 78%, $p < 0.0001$). Soluble PD-L1 suppression predicted overall survival. The overall response rate was 61% (16.7% complete, 44.4% partial, 22.2% stable, and 16.7% progressing). Changes in peripheral immune cell populations and immunosuppressive factors predicted overall survival. sPD-L1 and other circulating immunoregulatory molecules mediate ICI resistance. TPE can reduce these factors and resensitize ICI-refractory melanoma. Patients with persistent elevation or rapid rebound of sPD-L1 experienced inferior outcomes, suggesting that multiple courses of TPE may be necessary. These findings may apply to other ICI-resistant cancers. Trial registration: NCT04581382, ReCIPE-M1 (Rescuing Cancer Immunotherapy with Plasma Exchange in Melanoma 1).

Programmed death-ligand 1 (PD-L1) on the surface of cancer cells engages receptor programmed cell death protein 1 (PD-1) on the surface of immune cells to prevent anti-tumor immunity[1]. Immune checkpoint inhibitor therapies (ICI) blocking PD-1/PD-L1 and other immunosuppressive ligand-receptor interactions are effective against many cancers, including melanoma[2]. Unfortunately, most cancers eventually develop ICI resistance.

Soluble immunosuppressive factors prevent the therapeutic effect of ICI. The most well-studied of these factors is soluble PD-L1 (sPD-L1). Produced by proteolytic cleavage or by alternative splicing,

sPD-L1 binds PD-1 directly on the surface of T cells to dampen anti-tumor immunity despite ICI treatment[3–6]. High plasma sPD-L1 predicts ICI resistance in multiple cancers[7–12]. Unsurprisingly, sPD-L1 impairs circulating tumor-reactive ($T_{TR}$) CD8+CD11a^high T cells that are critical for ICI response[13–15]. Similar properties are shared by other cancer-derived soluble factors, including B7H3 (CD276, PD-L3), CTLA-4 (CD152), Lymphocyte-activation gene 3 (LAG3), PD-L2 (PDCD1LG2), and T-cell immunoglobulin and mucin domain protein 3 (TIM3)[16–21]. Although their mechanisms and contributions to tumor cell immune checkpoint evasion vary, each of these soluble factors has the potential

✉e-mail: orme.jacob@mayo.edu

to overwhelm ICI therapy and prevent anti-tumor immunity in cancer. There is a critical need to address this Gordian knot of soluble factors mediating immunosuppression and ICI resistance in cancer.

Therapeutic plasma exchange (TPE) is a safe and widely available clinical procedure in which patient blood is filtered or centrifuged to remove deleterious substances. We previously demonstrated that TPE reduces sPD-L1 from circulation[22]. In other studies, we have demonstrated that stereotactic body radiotherapy (SBRT) to metastatic sites of disease can expand critical T cell populations that drive ICI response[23-25]. Based on these observations, we proposed the clinical use of SBRT and TPE to re-sensitize ICI-refractory cancers.

In the present study (NCT04581382), we explored sequential limited SBRT, TPE, and ICI re-challenge in the setting of ICI-refractory metastatic melanoma with high (≥1.7 ng/mL) sPD-L1 (Supplementary Fig. 1).

## Results

### Patients and baseline characteristics
Thirty-four (34) patients with metastatic melanoma progressing on anti-PD-1 ICI were screened for eligibility from December 2020 to February 2023 (Supplementary Figs. 1, 2). Sixteen (16) patients were excluded due to baseline sPD-L1 < 1.7 ng/mL ($n = 8$), poor venous access ($n = 2$), unwillingness to undergo treatment or follow-up ($n = 2$), contraindications to ICI ($n = 2$), or death before sPD-L1 level could be assessed ($n = 2$).

Eighteen (18) patients were eligible (Table 1). Mean age of patients in the study was 63 years (interquartile range [IQR] 12.5), and 7 (39%) were female (self-reported). Mean baseline sPD-L1 measured by ELISA was 26.31 ng/mL (IQR 18.57 ng/mL). All patients had previously received at least one PD-1 inhibitor that had progressed by RECIST criteria, and fifteen (83%) had previously received at least one ICI doublet (e.g., ipilimumab and nivolumab). Half had received two prior lines of therapy, and an additional one third of patients had received four or more lines of therapy before enrollment in the study. Sixteen patients (89%) were on current failing ICI therapy at enrollment, while two patients (11%) were on an alternate failing therapy at the time of enrollment but had at least one prior failed ICI therapy.

Treatment comprised 1–5 days of stereotactic body radiotherapy (SBRT) to a minority of known sites of disease (Table 2, Supplementary Table 1). A median of 1.5 lesions (mean 1.8, IQR 1.0) per patient were irradiated, and a median 9.5 lesions (mean 12.7, IQR 9.25) were not irradiated (non-irradiated) for each patient. The maximum number of lesions irradiated in any patient was 4, and the maximum percent of lesions irradiated in any patient was 33%. The median radiation exposure for each irradiated lesion was 40 Gy in 5 fractions. SBRT was followed by three consecutive daily sessions of TPE and re-challenge with ICI. One patient was unable to complete the final TPE treatment due to a line infection. All patients received ICI re-challenge, and sixteen patients (89%) received the same anti-PD-1 ICI they had received in at least one prior line of therapy, while two patients (11%) received a new anti-PD-1 ICI monotherapy. Fifteen patients (83%) received nivolumab. All patients were included in the intent-to-treat analysis regardless of treatment completion. Blood samples were taken at registration before SBRT (baseline), after SBRT and before TPE (pre-TPE), after TPE and before ICI re-challenge (post-TPE), and before the second cycle of ICI re-challenge (ICI2).

### Primary safety and efficacy outcomes
The primary feasibility endpoint of this trial was to assess the adverse events (AE) observed during SBRT, TPE, and ICI re-challenge in patients with melanoma receiving PD-1 immunotherapy (Table 3). AEs were collected and reported according to CTCAE v.5.0. In a pre-planned stopping rule, if 3 of the first 10 patients were to experience grade 4 or greater adverse events attributable to the intervention, the study would be halted.

In total, one patient experienced a thromboembolic event (grade 3) and sepsis (grade 4) secondary to an infection of a vascular access device. This patient required hospitalization, line removal, and intravenous antibiotics. This patient also experienced grade 2 colitis attributable to ICI therapy, leading to discontinuation. Other grade 3 AE were fatigue and malaise. No deaths were attributable to therapy. All patients were enrolled in the expected timeline.

The primary efficacy endpoint of the study was the reduction of sPD-L1 levels by TPE (Fig. 1A, Supplementary Fig. 3). Levels of sPD-L1 were significantly reduced by TPE (mean 78% reduction, $p = 3 \times 10^{-5}$). Median recovery rate of sPD-L1 before the second cycle of ICI re-challenge (ICI2) was 60% of pre-TPE levels ($p = 5.8 \times 10^{-4}$).

### Secondary efficacy outcomes
A secondary endpoint of the study was overall survival (OS). Median OS was 17.4 months (95% CI 14.7-NR) with a median follow-up of 21.2 months (Supplementary Fig. 4A). We sought to determine whether the extent of sPD-L1 recovery prior to cycle 2 of ICI-rechallenge predicted survival (Fig. 1B). Patients whose sPD-L1 levels remained below pre-TPE levels at ICI2 experienced superior OS relative to those with complete sPD-L1 recovery (HR 0.1 [95% CI 0.02-0.51], log rank $p = 0.0009$). This difference was more pronounced in patients with the highest sPD-L1 recovery at ICI2 (Supplementary Fig. 4B). Baseline sPD-L1 prior to treatment did not predict OS ($p = 0.47$).

A further secondary efficacy endpoint was overall response rate (ORR), the proportion of patients experiencing CR or PR versus SD or PD per RECIST v1.1. Only non-irradiated lesions were used to determine response (i.e., irradiated lesions were excluded from analysis per RECIST standards). The ORR was 61%. Three patients (16.7%) experienced CR, eight (44.4%) PR, four (22.2%) SD, and three (16.7%) PD as best response (Table 4). Absolute post-TPE sPD-L1 levels were lower in patients who experienced CR ($p = 0.046$, Supplementary Fig. 5A). Post-TPE sPD-L1 level comparisons for responders (i.e., CR and PR) versus non-responders (i.e., SD and PD) were not statistically significant (Supplementary Fig. 5B).

Representative irradiated and non-irradiated lesions from each patient were assessed by a blinded radiologist for standardized uptake value (SUV) and tumor size over time. 78% of irradiated and 54% of non-irradiated lesions showed a greater than 50% reduction in SUV (Fig. 1C). 91% of measured irradiated and 70% of non-irradiated lesions reduced in size (Fig. 1D). Example FDG PET scan images of a patient experiencing CR is shown in Fig. 1E. A swimmer plot of all patients by prior ICI response and resistance, ICI regimen, TPE completion, BRAF mutation status, and response to ICI challenge is shown in Fig. 1F. Patients with a prior secondary resistance to ICI experienced superior OS in comparison to patients with prior primary ICI resistance (HR 0.28 [95% CI 0.08-0.97], $p = 0.04$).

Median progression-free survival (PFS) for the cohort was 3.9 months (95% CI 3.0-7.4) with a median PFS follow-up of 22.2 months. Strong reduction in sPD-L1 by TPE predicted superior PFS (HR 0.21 [95% CI 0.05-0.79] $p = 0.02$) and surging sPD-L1 at ICI2 predicted inferior PFS (HR 3.28 [95% CI 1.04-10.3], $p = 0.03$), while sPD-L1 recovery at ICI2 trended toward inferior PFS (HR 2.53 [95% CI 0.87-7.34], $p = 0.08$). Baseline sPD-L1 did not predict PFS ($p = 0.74$). In an exploratory analysis, we measured the duration of response (DOR). The mean DOR was 7.7 months (median 4.6 months, [95% CI 3.3-NR]).

### Tumor-reactive effector T cell changes associate with outcomes to immunotherapy re-challenge after radiotherapy and TPE
Peripheral blood mononuclear cell (PBMC) subpopulations are liquid biomarkers and critical substrates of ICI immunotherapy. Changes in these subpopulations reflect the effects of therapy on anti-tumor immunity.

We isolated PBMCs and measured subpopulations by flow cytometry (Supplementary Tables 2, 3, Supplementary Figs. 6, 7). We

**Table 1 | Baseline patient characteristics and plasma sPD-L1**

| Table of patient characteristics | | |
|---|---|---|
| | | $N = 18$[1] |
| Gender | | |
| | Male | 11/18 (61%) |
| | Female | 7/18 (39%) |
| Race | | |
| | White | 18/18 (100%) |
| Ethnicity | | |
| | Non-Hispanic | 18/18 (100%) |
| Number of Prior Lines of Treatment | | |
| | 1 | 2/18 (11%) |
| | 2 | 9/18 (50%) |
| | 3 | 1/18 (6%) |
| | 4+ | 6/18 (33%) |
| Age | | 63.2 (13.2) |
| LDH at first recurrence or metastatics diagnosis in (U/L) | | 331.9 (327.6) |
| ECOG status | 0 | 12 (66.7%) |
| | 1 | 5 (27.8%) |
| | 2+ | 1 (5.6%) |
| sPD-L1 in (ng/mL) | | 26.31 (40.01) |
| Histology | Nodular melanoma | 7 (38.9%) |
| | Unspecified subtype | 7 (38.9%) |
| | Other (acral, mucosal in the anal region, or uveal) | 4 (22.2%) |
| BRAF mutation status | Wild-type | 12/18 (66.7%) |
| | V600E | 3/18 (16.7%) |
| | V600R | 1/18 (5.6%) |
| | K601E | 2/18 (11.1%) |
| Prior ICI received[2] | Atezolizumab | 1 (5.6%) |
| | Ipilimumab | 14 (77.8%) |
| | Nivolumab | 17 (94.4%) |
| | Pembrolizumab | 10 (55.6%) |
| | Relatlimab | 2 (11.1%) |

ECOG is the Eastern Cooperative Oncology Group performance status, ranging from 0 (fully active) to 5 (dead). Tumor types are classified according to the primary site of origin and histology. Individual patient-level data are reported in Supplementary Data 1[29]. *ICI* Immune checkpoint inhibitor, *LDH* lactate dehydrogenase.
[1]Summarized as n/N (%) or mean (sd).
[2]15 patients (83%) previously received at least one ICI doublet therapy.

**Table 2 | Treatment characteristics. Individual patient-level data are reported in Supplementary Data 1**

| Table of treatment characteristics | | |
|---|---|---|
| Lesions per patient[1] | | |
| | Irradiated lesions | 1.5 (IQR 1) |
| | Non-irradiated | 9.5 (IQR 9.25) |
| Percent of lesions irradiated per patient | | |
| | Median | 12.1% |
| | Mean | 14.7% |
| | Minimum | 6.25% |
| | Maximum | 33.3% |
| Number of completed TPE[2] | | |
| | 2 sessions | 1/18 (5.6%) |
| | 3 sessions | 17/18 (94.4%) |
| Re-challenge ICI regimen[2] | | |
| | Nivolumab monotherapy | 15/18 (83.3%) |
| | Pembrolizumab monotherapy | 3/18 (16.7%) |

[1]Summarized as median (IQR).
[2]Summarized as n/N (%) or mean (sd). All re-challenge ICI regimens were ICI monotherapy. *ICI* Immune checkpoint inhibitor.

**Table 3 | Adverse events on treatment**

| Adverse Event | Grade | | | | |
|---|---|---|---|---|---|
| | 1 | 2 | 3 | 4 | 5 |
| Type | N (%) | N (%) | N (%) | N (%) | N (%) |
| **Fatigue** | 6 (38%) | 4 (25%) | 1 (6%) | | |
| **Arthralgia** | 6 (38%) | | | | |
| **Rash maculo-papular** | 5 (31%) | | | | |
| **Malaise** | 2 (12%) | | 1 (6%) | | |
| **Diarrhea** | 2 (12%) | | | | |
| **Hypotension** | 1 (6%) | 1 (6%) | | | |
| **Myalgia** | 2 (12%) | | | | |
| **Alopecia** | | 1 (6%) | | | |
| **LFT abnormality** | 1 (6%) | | | | |
| **Paresthesia** | 1 (6%) | | | | |
| **Pneumonitis** | | 1 (6%) | | | |
| **Sepsis** | | | | 1 (6%) | |
| **Thromboembolic event** | | | 1 (6%) | | |

One patient experienced a grade 4 sepsis and grade 3 thromboembolic event related to a vascular access line that required hospitalization, line removal, and intravenous antibiotics. Another patient experienced grade 3 fatigue and malaise. All other AEs were grade 1-2. *LFT* liver function test.

compared subpopulations from baseline to ICI2 in each patient (Fig. 2A). We further evaluated how these changes correlated with OS in our cohort by Cox proportional hazards (Fig. 2B, Supplementary Tables 4, 5).

Our group previously identified tumor-reactive CD8+ T cells ($T_{TR}$) with high expression of CD11a, Granzyme B, and CX3CR1 (GZMB+/CX3CR1+/CD11a^high or $T_{TR}$)[23,24,26]. In a prespecified analysis, peripheral blood $T_{TR}$ changes from baseline to ICI2 predicted overall survival (OS) with a Cox proportional hazard ratio (HR) of 0.15 (95% confidence interval [CI] 95% CI 0.04-0.6, $p = 0.008$; Fig. 2C, log rank $p = 0.002$). Similar results were observed when $T_{TR}$ levels were analyzed dichotomously as increasing versus decreasing. A combination of low sPD-L1 after TPE and increasing $T_{TR}$ strongly predicted improved OS ($p = 0.003$, Supplementary Fig. 8).

Regulatory T cells marked by CD4, CD25, and FOXP3 expression ($T_{reg}$) are well-established predictors of poor response to immunotherapy. In a prespecified analysis, higher peripheral blood $T_{reg}$ from baseline to ICI2 was associated with poor OS (HR 9.72 [95% CI 2-47.3], $p = 0.005$; Fig. 2D, log rank $p = 0.0009$).

We previously showed that NKG7 marks resilient T cells that are critical to ICI response and, conversely, that Bim (BCL-2-interacting mediator of cell death) is upregulated in T cells by PD-L1/PD-1 engagement and drives poor anti-tumor immunity[23,27,28]. In an exploratory analysis, we measured changes in peripheral blood CD11a^high CD8 + T cells with NKG7 or Bim expression from baseline to the next cycle of ICI re-challenge (Fig. 2E, F). Increasing NKG7 trended toward superior OS (HR 0.32 [95% CI 0.1-1.07], $p = 0.064$) and increasing Bim predicted inferior OS (HR 5.5 [95% CI 1.4-21], $p = 0.014$).

**Exploratory soluble biomarkers and outcomes**

Soluble factors beyond sPD-L1 contribute to ICI resistance. We next sought to determine the effect of TPE on these soluble factors in

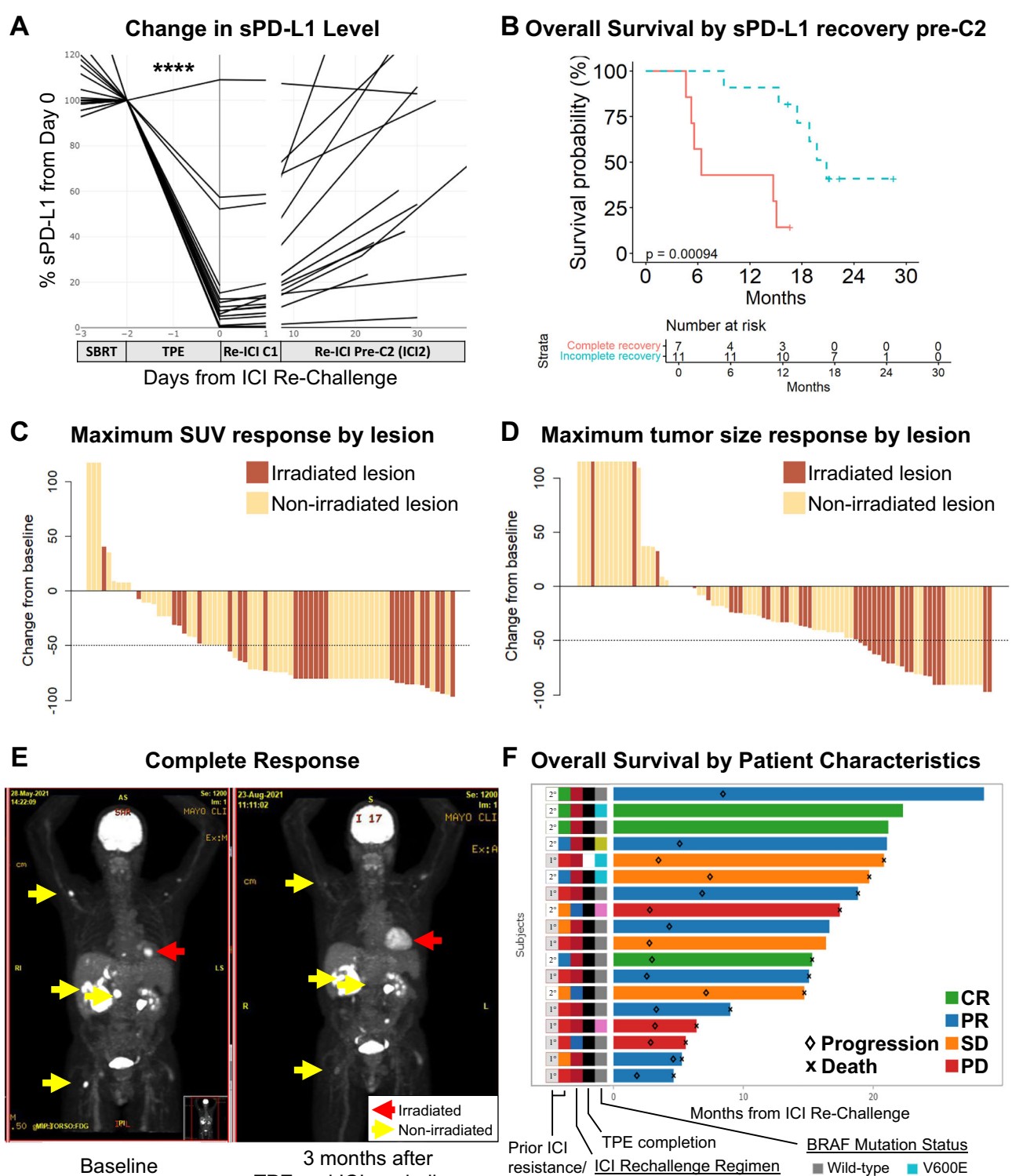

**A** Change in sPD-L1 Level

**B** Overall Survival by sPD-L1 recovery pre-C2

**C** Maximum SUV response by lesion

**D** Maximum tumor size response by lesion

**E** Complete Response

Baseline　　　　　3 months after TPE and ICI re-challenge

**F** Overall Survival by Patient Characteristics

systemic circulation. We measured a large library of soluble factors in plasma samples from patients on the study and compared these to matched healthy controls using a high-throughput O-link multiplex assay. A variety of immunosuppressive soluble factors trended toward elevation at baseline in the blood of patients with ICI-refractory melanoma versus healthy controls (Fig. 3A, Supplementary Data 1)[29], including sCTLA-4, sLAG3, and sTIM-3. As an internal control, sPD-L1 was reduced by TPE and low post-TPE sPD-L1 predicted superior OS (Fig. 3B).

We compared these factors in the plasma of patients from SBRT to TPE, pre- to post-TPE, and at the second cycle of ICI re-challenge (Fig. 3C, F; Supplementary Fig. 9). TPE significantly reduced sPD-L2, sTIM3 (sHAVCR2), sB7H3, and sLAG3 but not CTLA4 in our cohort. Post-TPE sPD-L2 and sTIM3 levels, like sPD-L1 levels, correlated with OS (Fig. 3D, G; $p = 0.002$ and 0.008, respectively). However, the levels of these factors at ICI2 were not predictive (Fig. 3E, H). Levels of sB7H3 prior to the second cycle of ICI re-challenge correlated with OS ($p = 0.015$).

**Fig. 1 | Treatment outcomes after SBRT, TPE, and ICI re-challenge. A** In the primary efficacy endpoint, TPE removed a mean 78% of sPD-L1 (Wilcoxon $p = 0.00003$). Median recovery was 60% of pre-TPE levels ($p = 0.00058$). Absolute sPD-L1 levels are shown in Supplementary Fig. 3. **B** OS is estimated in patients who experienced complete recovery of sPD-L1 before cycle 2 of ICI re-challenge (complete recovery) and those with sPD-L1 that remained suppressed at cycle 2 of ICI rechallenge (incomplete recovery). Patients with incomplete sPD-L1 recovery experienced superior OS (log rank $p = 0.0009$, HR 0.1 [95% CI 0.02-0.51]). **C** A blinded radiologist assessed representative lesions among all patients (at least two lesions per patient) for SUV change on treatment. SUV for each lesion at each timepoint was calculated by mean blood pool SUV. Irradiated and non-irradiated index lesions are plotted. 63% of lesions had an SUV reduction greater than 50% (78% of irradiated lesions, 54% of non-irradiated lesions). **D** A blinded radiologist assessed representative lesions among all patients (at least two lesions per patient) for size change on treatment. Irradiated and non-irradiated index lesions are plotted. 78% of lesions reduced in size (91% of irradiated lesions, 70% of non-irradiated lesions) on treatment. **E** An example of one patient with CR after irradiation of one lesion, TPE, and ICI re-challenge. Non-irradiated lesions are also marked. **F** Patient survival over time is plotted by treatment response. Oncoprint shows prior ICI response (CR, PR, SD, or PD), prior ICI resistance type (secondary versus primary), ICI re-challenge regimen (nivolumab versus pembrolizumab), and BRAF mutation status for each patient. Swimmer plot is colored by ICI re-challenge response (CR, PR, SD, or PD). Patients marked with "◇" experienced progression at the timepoint specified. Patients marked with "x" died at the timepoint specified. TPE therapeutic plasma exchange, SBRT stereotactic body radiotherapy, ICI immune checkpoint inhibitor, OS overall survival, HR hazard ratio. CR complete response, PR partial response, SD stable disease, PD progressing disease, SUV standardized uptake value (SUV). All statistical tests were two-sided. See also Supplementary Figs. 3-5.

## Table 4 | Response rates

| Overall | |
|---|---|
| n | 18 |
| Response (%) | |
| CR | 3 (16.7) |
| PR | 8 (44.4) |
| SD | 4 (22.2) |
| PD | 3 (16.7) |

Overall best response was assessed. *CR* complete response, defined as the absence of radiologically apparent disease. *PR* partial response, defined as greater than 30% radiographic reduction. *SD* stable disease, defined as less than 30% radiographic reduction. *PD* progressing disease, defined as an increase of any lesion by 20%. Only non-irradiated lesions were used to determine response (i.e., irradiated lesions were excluded from analysis per RECIST standards).

## Discussion

Soluble immunosuppressive factors present a significant challenge to ICI therapy. While PD-1 inhibitors and sPD-L1 are the best studied classes of ICI and resistance-inducing soluble immunosuppressive factors, respectively, they are archetypal of other checkpoint interactions. Current ICI therapies may be overwhelmed by the volume and variety of these factors in the blood of patients with ICI-resistant malignancies. TPE may offer a method to remove these factors and restore ICI sensitivity.

This study met its primary safety and efficacy endpoints. AEs were commensurate with other studies of immunotherapy. TPE effectively reduced plasma sPD-L1, although the level of sPD-L1 and other soluble factors known to limit anti-tumor immunity rebounded rapidly prior to the second cycle of ICI.

This study presents clinical evidence that a combination of limited SBRT, TPE, and ICI re-challenge may restore ICI responsiveness. We observed an ORR of 61%. This ORR is remarkable for PD-1 ICI monotherapy in such a heavily pre-treated population, the majority of which had previously received multiple lines of therapy, including ICI doublet therapy, and all of whom were experiencing disease progression a the time of enrollment. 89% of patients received the same PD-1 ICI that had failed in a prior line of therapy.

A recent SWOG study provides important context to our findings and an alternative approach to ICI resistance (S1616, NCT03033576)[30]. In that study, patients with anti-PD-1 ICI-refractory melanoma were randomized to anti-CTLA-4 ipilimumab monotherapy or the ipilimumab and nivolumab doublet in the second line. Anti-CTLA-4 ICI monotherapy achieved a 9% response rate (0% CR) as compared with 28% in the doublet arm. A small minority of these patients had previously received an ICI doublet. We anticipate that SBRT and doublet ICI would result in a similar response rate. Similarly, a recent study of ICI restart after adjuvant anti-PD-1 therapy showed no responses for patients with on-therapy progression and an approximate 40% response rate after progression off of therapy[31]. It is unknown whether a combination of multiple ICI modalities and SBRT or TPE could further improve responses. The present study provides provocative correlative markers that may help inform future approaches. First, treatment induced significant changes in peripheral circulating immune cells. Some of these changes, including tumor reactive CD8+ T cells that express CD11, Granzyme B, and CX3CR1 and classical regulatory T cells, predicted clinical outcomes despite the limited number of patients in the study. A favorable shift in the balance of anti-tumor immunity and tolerance provides evidence that ICI-resistant tumors remain potentially sensitive to immunotherapy. Furthermore, these changes reflect the accepted mechanism of treatment response and may help guide future studies as biomarkers. Second, TPE significantly reduced an array of soluble mediators of immunosuppression. The reduction of several of these factors also predicted clinical outcomes in this cohort. While soluble mediators of ICI resistance will vary widely from patient to patient and from cancer to cancer, TPE offers a broad way to address many such mediators. In all, we posit that SBRT and TPE may be used to "reset" systemic immunity and restore ICI sensitivity.

Several limitations to this study should be acknowledged. First, this study combined three sequential treatments—SBRT, TPE, and PD-1 ICI re-challenge—in an open-label, single-arm trial. While biomarkers were measured at multiple time points in the study, the individual contribution of SBRT and TPE to overcoming ICI resistance is uncertain. Although SBRT was highly effective against individual metastases, most lesions were not irradiated. Of these non-irradiated lesions, most showed a reduction in both size and FDG avidity. However, these observations must not be taken as evidence for a purported "abscopal" effect—meaning involution of non-irradiated tumors after radiation of other metastases—because this trial included neither an SBRT-free control nor a TPE-free comparator arm from which to draw such conclusions. Such a study would help determine the relative contribution of each intervention, which is not possible in the present study. Our previous studies of SBRT alone did not show significant short-term changes in sPD-L1[24], but direct tumor killing would be expected to reduce sPD-L1 production. In contrast, our preliminary study showed that TPE directly removes sPD-L1 from circulation[22]. Thus, a potential synergy of SBRT and TPE is that TPE may eliminate circulating factors and SBRT may reduce the long-term de novo production of these factors from tumors. Additional study is needed.

Why would a transient decrease in circulating sPD-L1, TIM3, or other factors lead to ICI response? We hypothesize that the reduction of these soluble factors, combined with immediate PD-1 antagonism, allows activation of key anti-tumor immune cells. The reduction in sPD-L1 and increase in circulating $T_{TR}$, for example, was greatest in those patients experiencing a response. A composite of both sPD-L1

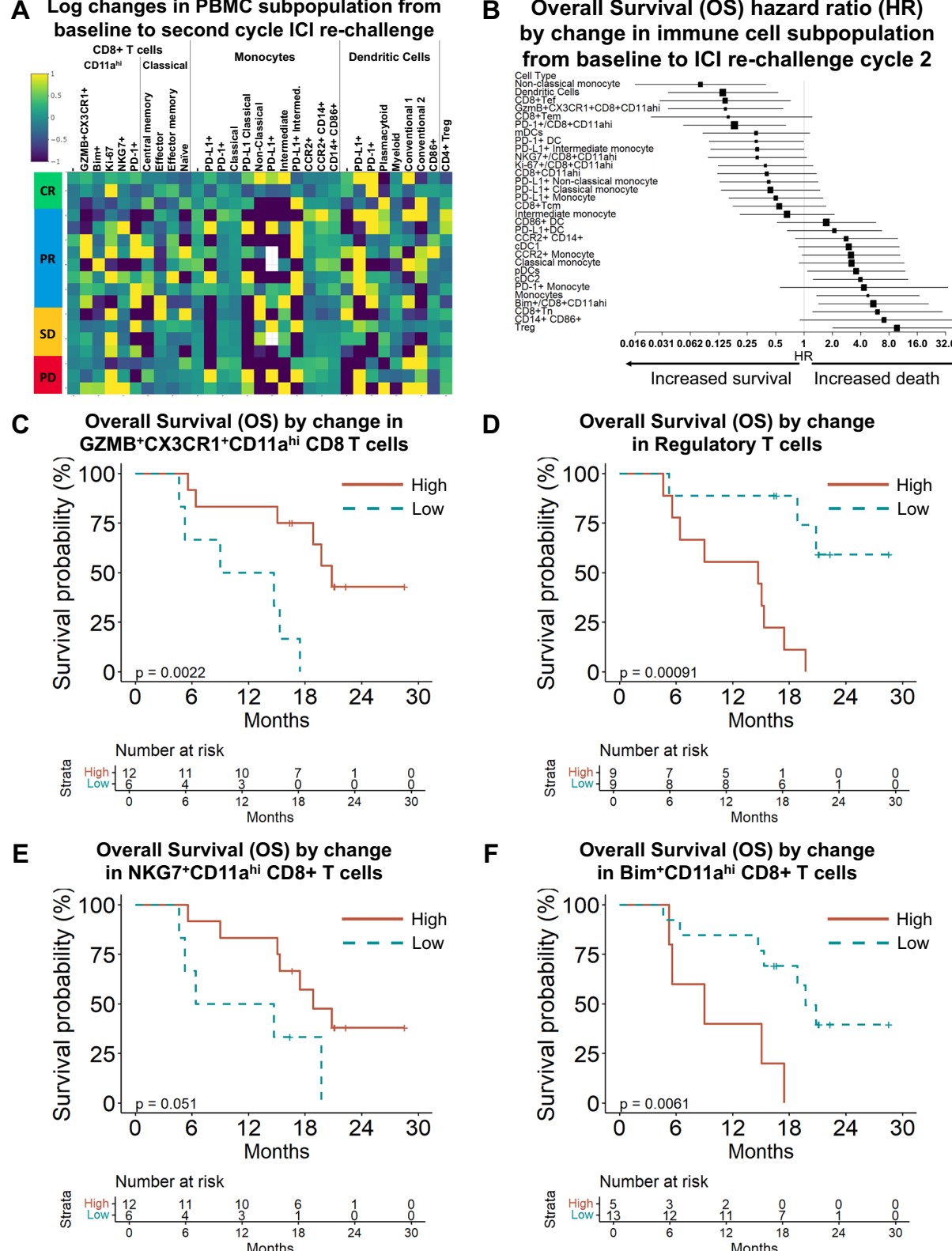

**A** Log changes in PBMC subpopulation from baseline to second cycle ICI re-challenge

**B** Overall Survival (OS) hazard ratio (HR) by change in immune cell subpopulation from baseline to ICI re-challenge cycle 2

**C** Overall Survival (OS) by change in GZMB⁺CX3CR1⁺CD11aʰⁱ CD8 T cells

**D** Overall Survival (OS) by change in Regulatory T cells

**E** Overall Survival (OS) by change in NKG7⁺CD11aʰⁱ CD8⁺ T cells

**F** Overall Survival (OS) by change in Bim⁺CD11aʰⁱ CD8⁺ T cells

reduction and $T_{TR}$ changes was a better predictor of OS than either variable alone.

While the clinical activity observed in this study is favorable and warrants further study, most patients experienced disease progression and death. Even in patients with a favorable response, there was re-accumulation of immunosuppressive factors in the bloodstream and a

mean DOR of 7.7 months. It is possible that a more sustained suppression of these factors (i.e., performing serial TPE prior to multiple ICI cycles) would improve the robustness of ICI re-challenge response. As a further limitation, this study did not include novel ICI classes. Thus, additional clinical trials are needed to determine how best to leverage this approach in ICI resistance.

**Fig. 2 | Tumor effector T cell changes predict outcomes to immunotherapy re-challenge after radiotherapy and TPE. A** PBMCs were isolated from each patient and subpopulations were measured by a broad flow cytometry panel and change in percent of each subpopulation is reported for each patient on a log scale. **B** Cox proportional hazards regression was performed for changes in each subpopulation. Hazard ratios with 95% confidence intervals are shown; box size indicates proportion of patients with increasing immune cell subpopulation. **C** Overall Survival (OS) is predicted by change in GZMB$^+$CX3CR1$^+$CD11a$^{high}$ CD8$^+$ (T$_{TR}$) T cells from baseline to the second cycle of ICI re-challenge (log rank $p$ = 0.002; HR 0.15 [95% CI 0.04-0.6], $p$ = 0.008). **D** Overall Survival (OS) is predicted by change in

CD25$^+$FOXP3$^+$ CD4$^+$ (T$_{reg}$) T cells from baseline to the second cycle of ICI re-challenge (log rank $p$ = 0.0009; HR 9.72 [95% CI 2-47.3], $p$ = 0.005). **E** Overall Survival (OS) by change in NKG7$^+$CD11a$^{high}$ CD8$^+$ T cells from baseline to the second cycle of ICI re-challenge is shown (log rank $p$ = 0.05; HR 0.32 [95% CI 0.1-1.07], $p$ = 0.064). **F** Overall Survival (OS) is predicted by change in Bim$^+$ T cells from baseline to the second cycle of ICI re-challenge (log rank $p$ = 0.006; HR 5.5 [95% CI 1.4-21], $p$ = 0.014). CI confidence interval, GZMB Granzyme B, HR Cox proportional hazard ratio, PBMCs peripheral blood mononuclear cells. All statistical tests were two-sided. See also Supplementary Tables 2–5, Supplementary Figs. 6, 7.

Taken together, these data outline a promising paradigm for overcoming ICI resistance. Future studies are warranted to validate and optimize this approach in melanoma and other malignancies.

## Methods

### Study design and participants
Study design and conduct of ReCIPE-M1 (Rescuing Cancer Immunotherapy with Plasma Exchange in Melanoma) complied with all relevant regulations regarding the use of human study participants and was conducted in accordance with the criteria set by the Declaration of Helsinki. We conducted the study under a protocol approved by the Mayo Clinic institutional review board and registered at ClinicalTrials.gov (NCT04581382). Full protocol and consent forms are in Supplementary Information. This was a prospective, open-label, single-arm phase I/II trial for patients with advanced ICI-resistant melanoma with elevated sPD-L1 levels. Written informed consent was obtained from all participants prior to study enrollment. All data were collected in a deidentified fashion in a password-protected database.

Inclusion criteria included age 18 or older, histologically confirmed melanoma, measurable disease per RECIST criteria, ECOG (Eastern Cooperative Oncology Group) status of 3 or better, sPD-L1 level 1.7 ng/mL or greater, and feasible vascular access. Exclusion criteria included contraindications to PD-1 inhibitor ICI, unwillingness to undergo treatment and follow-up, or active biotin supplementation due to the known interference of biotin supplements with accurate measurement of sPD-L1[5]. Patients were not compensated for study participation.

**Recruitment.** Patients with metastatic melanoma progressing despite ICI treatment were approached at a single academic center between December 2020 and February 2023. Patients were screened for sPD-L1 levels using a sandwich enzyme-linked immunosorbent assay (ELISA) kit (see Biomarker Assays). Patients who met the inclusion and exclusion criteria were considered positive and eligible for the study.

**Treatment.** Consenting and eligible patients underwent 1 to 5 days of SBRT to a minority of metastatic sites seen on imaging (Table 1, Supplementary Table 1). Patients next underwent three daily sessions of TPE, performed using centrifugation-based cell separators, either the Fenwal Amicus (Fresenius KABI USA LLC, Lake Zurich, Illinois, USA) or the Spectra Optia (Terumo BCT, Lakewood, Colorado, USA). For each patient, a single plasma volume was exchanged at each session and replaced with albumin containing no sPD-L1. Vascular access lines were removed after the last planned TPE session. After completing TPE, patients received anti-PD-1 ICI monotherapy re-challenge per clinician preference. ICI cycle duration was determined per product label as a standard of care.

**Assessments.** Clinical standard of care imaging was performed per clinician preference, generally comprising conventional computed tomography (CT) imaging with or without advanced molecular [$^{18}$F] Fluorodeoxyglucose positron emission tomography (FDG PET) imaging every 3 months. Changes in size were reported as best response

per Response Evaluation Criteria in Solid Tumors (RECIST) guidelines version 1.1[32]. Per the criteria, only non-irradiated lesions were used to determine response. Complete response (CR) was defined as resolution of radiologically apparent disease on conventional imaging, partial response (PR) as greater than 30% radiographic reduction on conventional imaging, stable disease (SD) as less than 30% radiographic reduction, and progressing disease (PD) as 20% or greater increase in radiographically apparent disease. PET avidity was not used to determine response. For waterfall plots, PET-avid lesions were included, and at least two representative metastatic sites (one irradiated and one non-irradiated) were reported for each patient.

### Study endpoints
The primary safety endpoint was feasibility as measured by the rate of adverse events (AE) under the revised National Cancer Institute Common Terminology Criteria for Adverse Events (CTCAE) version 5.0. The maximum grade for each type of AE was recorded for each patient.

The primary efficacy endpoint was the removal of sPD-L1 after plasma exchange. Secondary efficacy endpoints included overall survival (OS) after TPE; overall response rate (ORR) defined as patients with complete response (CR) or partial response (PR); and progression-free survival (PFS) after TPE. Correlative analyses comprised blood mononuclear cell (PBMC) subpopulations, including prespecified T cell population changes and plasma proteomic analysis as outlined below.

### Biomarker assays
**Soluble PD-L1 (sPD-L1).** sPD-L1 was measured in a blinded manner by enzyme-linked immunosorbent assay (ELISA) as previously published[5]. This assay comprises paired mouse IgG2 monoclonal antibody clones H1A (capture) and biotinylated B11 (detection) against extracellular human PD-L1 using a standard streptavidin-HRP method. Concentrations were determined by optical density (OD) measurements along a known standard curve of recombinant human PD-L1. Assays were performed in a blinded fashion in triplicate and reported in nanograms per milliliter (ng/mL).

**Olink proteomic profiling.** Soluble plasma markers were assessed in a blinded manner using a commercially available platform (Olink) and compared to age-matched healthy controls. Plasma samples were labeled with oligonucleotide-labeled antibody probes and subjected to microfluidic PCR amplification using a Proximity Extension Assay (PEA) with NGS readout on Illumina instruments. Assays were performed in a blinded fashion in triplicate and reported in relative units on a normalized scale.

**Peripheral blood mononuclear cell (PBMC) profiling.** Peripheral blood mononuclear cells (PBMC) were profiled for established cell types in a blinded manner (see Supplementary Tables 2, 3). We stained surface markers prior to intracellular markers. Flow cytometry data were collected on a Cytek Aurora (Cytek Biosciences) and analyzed with FlowJo 10.10.0 (Tree Star). Changes in percent from baseline to next cycle of ICI re-challenge were compared and plotted. A

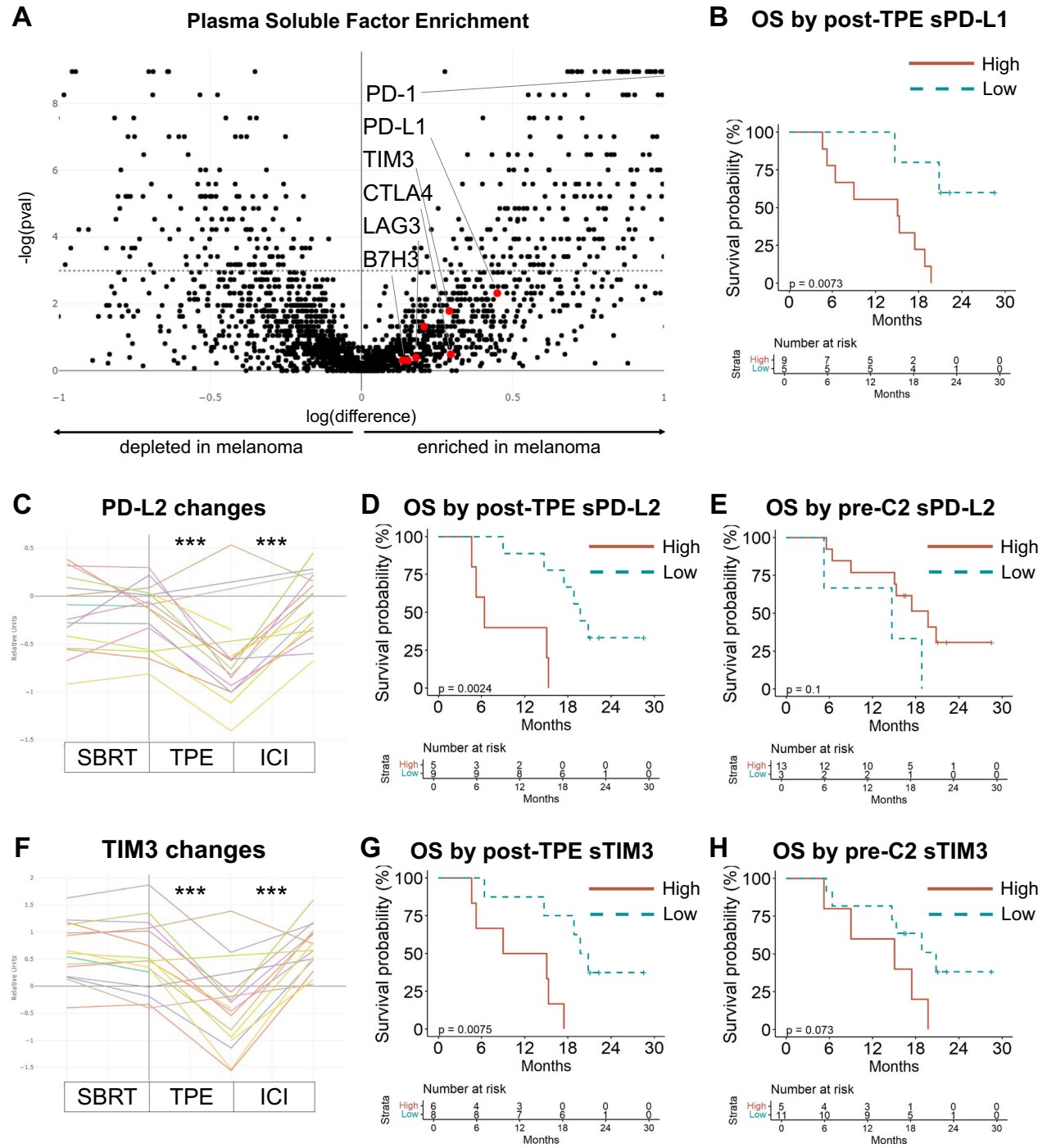

**Fig. 3 | Soluble mediators of immunosuppression are elevated in patients with ICI-refractory melanoma, are reduced by TPE, and reaccumulate. A** Soluble factors at baseline in patients with ICI-refractory melanoma and matched healthy controls were measured by O-link multiplex assay. The log difference in mean level (x axis) and two-sided *p* value of the difference by Student's *t* test (y axis) is shown for each soluble factor. **B** Post-TPE levels of sPD-L1 predicted OS (log rank *p* = 0.007). **C**–**E** sPD-L2 was significantly reduced (Wilcoxon *p* = 0.0006) by TPE and recovered (*p* = 0.0004) before the second cycle of re-challenge ICI. Post-TPE levels of sPD-L1 predicted OS (log rank *p* = 0.002) while pre-cycle 2 sPD-L2 did not (*p* = 0.1). **F**–**H** Soluble TIM3 was significantly reduced by TPE (Wilcoxon *p* = 0.0002) and recovered (*p* = 0.0004) before the second cycle of re-challenge ICI. Post-TPE levels of TIM3 predicted OS (log rank *p* = 0.008) while pre-cycle 2 TIM3 did not (*p* = 0.07). All statistical tests were two-sided, no adjustments were made for multiple comparisons. See Supplementary Fig. 8, Supplementary Data 1. C2 cycle 2, OS overall survival, TPE therapeutic plasma exchange.

prespecified analysis was performed for Granzyme B-positive, CX3CR1-positive, CD11a-high CD8-positive (GZMB[+]/CX3CR1[+]/CD11a[high] or T$_{TR}$) CD8[+] tumor-reactive T cells and CD4-positive, CD25-positive, FOXP3-positive (T$_{reg}$) regulatory T cells that are associated with response to immunotherapy[23,24,26]. Additional exploratory analyses were performed for each cell subtype.

## Statistical analysis

No formal prospective power calculation was performed due to the low expected response rate to ICI re-challenge. A total of 17 evaluable patients (up to 20) was estimated to be needed to assess primary efficacy and safety endpoints. In a post-hoc power analysis, a delta 4.46 (77.6% reduction) in sPD-L1 would be detected at 100% power in this

study. Conversely, powering at 90% would require a minimum of three patients. All patients who were eligible were included in the intention to treat (ITT) analysis regardless of treatment completion. OS was estimated by the Kaplan-Meier method and groups were further compared by Cox proportional hazards or log-rank tests as indicated. AE were reported by maximum grade in tabular form. Wilcoxon signed-rank test was used to assess changes in levels over time across the different timepoints of interest. Student's $t$ test was used for additional comparisons as indicated.

## Reporting summary
Further information on research design is available in the Nature Portfolio Reporting Summary linked to this article.

## Data availability
All raw data, including redacted individual patient data, and a data dictionary defining each field is available in Supplementary Data 1 (https://doi.org/10.6084/m9.figshare.27317040)[29]. Data underlying each figure are separately available in Source Data File 1. Study protocol and informed consent form are available in the supplementary information. Source data are provided with this paper.

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

## Acknowledgements
We thank the brave and selfless patients and their families who agreed to participate in this clinical trial. We thank clinical staff and study coordinators. Statistical guidance was provided generously by Nathan Foster of the Mayo Clinic Center for Clinical and Translational Science (CCaTS). We thank the Lawrence W. and Marilyn W. Matteson family for their generous support.

## Author contributions
J.J.O.: Conceptualization, Methodology, Writing, Funding Acquisition, Supervision. H.Z.: Formal Analysis, Investigation. P.L.: Formal Analysis, Investigation. Y.K.: Formal Analysis, Investigation. R.L.: Formal Analysis, Investigation. M.D.: Data Curation, Statistical Analysis. P.D.: Data Curation, Statistical Analysis. J.H.: Formal Analysis, Investigation. E.A.B.: Data Curation, Investigation. J.K.G.: Data Curation, Investigation. M.H.: Data Curation, Investigation. H.Dale: Project Administration, Investigation. DSS: Investigation. L.A.K.: Investigation. R.R.M.: Investigation. M.S.B.: Investigation. A.S.M.: Investigation. S.N.M.: Investigation. K.O.: Investigation. D.O.: Investigation. S.L.: Investigation. D.M.: Investigation. R.S.D.: Investigation. H.Dong: Data Curation, Investigation. F.L.: Data Curation, Investigation. A.T.P.: Data Curation, Investigation. J.L.W.: Data Curation, Methodology, Investigation. S.S.P.: Conceptualization, Methodology, Writing, Funding Acquisition, Supervision

## Funding
Funding for the study was provided by the Lawrence W. And Marilyn W. Matteson Fund at Mayo Clinic and the National Cancer Institute (R21-CA259236, JJO and SSP; K12-CA90628-23, JJO). Study sponsors were not involved in the study design, data collection, or analysis.

## Competing interests
Intellectual property has been filed, addressing discoveries disclosed in this manuscript. J.J.O. reports support and/or research and consulting support from the National Cancer Institute (R21 CA259236, K12 CA90628-23), Prostate Cancer Foundation, Mayo Lawrence W. and Marilyn W. Matteson Fund, NaNotics LLC, and Partner Therapeutics. L.K. reports consultant activities for Immunocore. M.S.B. reports institutional research support from Alkermes, Bristol-Myers Squibb, Genentech, Merck, nFerence, Perspective Therapeutics, Pharmacyclics, Regeneron, Sorrento, TILT Biotherapeutics, Transgene, and Viewpoint Molecular Therapeutics; as well as consultant activities for Sorrento Therapeutics, TILT Biotherapeutics, and Viewpoint Molecular Targeting; and receipt of drugs from Merck (pembrolizumab for an investigator-sponsored trial). A.S.M. reports grant funding from Novartis and Verily; support from Answers in CME, Antoni van Leeuwenhoek Kanker Instituut, AXIS Medical Education Inc, BeiGene, Chugai Pharmaceutical Co Ltd, Ideology Health LLC, Immunocore, Intellisphere LLC, Janssen, Johnson & Johnson Global Services, MJH Life Sciences, University of Miami Int'l Mesothelioma Symposium; consulting activities with AbbVie, AstraZeneca, BMS, Genentech, Gilead, Janssen, Sanofi Genzyme, Rising Tide, Takeda Oncology, and TRIPTYCH Health Partners; study funding subsequent publication processing fees from Genentech, Janssen, BMS; steering committee member for Janssen and Johnson & Johnson Global Services; travel support from Roche; was a non-remunerated director of the Mesothelioma Applied Research Foundation and is a non-remunerated director of the Friends of Patan Hospital. D.S.C. reports honoraria from Targeted Oncology, GU Onc Live, Curio, MJH Life Sciences, IntrinsiQ, International Centers for Precision Oncology Foundation; consulting or advisory roles in Janssen Biotech (Inst), Novartis (Inst), Abdera (Inst); research funding from Janssen Biotech (Inst), Novartis (Inst). R.R.M. reports grant funding from GlaxoSmithKline and BMS. S.N.M. reports grant funding, clinical trial for BMS, and grant funding, as well as PI for Journey Therapeutics and Sorrento Pharma. M.D. reports personal stock of less than $5,000 in Pfizer. K.O. reports an honorarium from UptoDate. D.O. reports research support from AstraZeneca and Varian; as well as an honorarium from UptoDate. S.P. reports support from NCI R21 CA259236 and the Mayo Lawrence W. and Marilyn W. Matteson Fund. D.M. reports intellectual property from EXACT Sciences, InSitu Biologics, and CurrentHealth. F.L. reports receiving research support tothe institution from Nanotics LLC; serving as scientific consultant for Mursla Bio; and receiving royalties from Early is Good. J.W. reports support from NCI R21 CA259236 and Mayo Lawrence W. and Marilyn W. Matteson Fund and non-remunerated director of AABB, ASFA, and ISFA, as well as an advisory board position for DSMB (paid). The remaining authors declare no competing interests.

## Additional information

Jacob J. Orme [1,2,9] ✉, Henan Zhang[3,9], Prashanth Lingamaneni[1], Yohan Kim[3], Roxane Lavoie[3], Maddy Dorr[4], Paul Dizona[4], Jacob Hirdler[3], Elizabeth A. Bering[2], Joanina K. Gicobi[1], Michelle Hsu[3], Heather Dale[5], Daniel S. Childs[1], Lisa A. Kottschade[1], Robert R. McWilliams [1], Matthew S. Block [1], Aaron S. Mansfield [1], Svetomir N. Markovic [1], Ken Olivier[6], Dawn Owen[6], Scott Lester[6], Daniel Ma[6], Roxana S. Dronca[7], Haidong Dong [3], Fabrice Lucien [3], Annie T. Packard[8], Jeffrey L. Winters [5] & Sean S. Park [6,9]

[1]Department of Oncology, Mayo Clinic, Rochester, MN, USA. [2]Department of Biochemistry and Molecular Biology, Mayo Clinic, Rochester, MN, USA. [3]Department of Immunology, Mayo Clinic, Rochester, MN, USA. [4]Department of Quantitative Health Sciences, Mayo Clinic, Rochester, MN, USA. [5]Department of Laboratory Medicine and Pathology, Mayo Clinic, Rochester, MN, USA. [6]Department of Radiation Oncology, Mayo Clinic, Rochester, MN, USA. [7]Department of Internal Medicine, Mayo Clinic, Jacksonville, FL, USA. [8]Department of Radiology, Mayo Clinic, Rochester, MN, USA. [9]These authors contributed equally: Jacob J. Orme, Henan Zhang, Sean S. Park. ✉e-mail: orme.jacob@mayo.edu

