## [Transparent Peer Review file · Nature Communications]

Plasma exchange and radiation resensitize immunotherapy-refractory melanoma: a phase I trial.

Corresponding Author: Dr Jacob Orme

Version 0:

Reviewer comments:

Reviewer #1

(Remarks to the Author)

Authors showed that RT+TPE might restore the sensitivity of ICI in patients with ICI-resistant melanoma. More than half of patients with advanced disease treated with ICI became resistant and have few choices of further treatment. I have several questions to authors as follows:

Major point:

1. Although the ORR seemed to be high (63%), the duration of response was short (median PFS <4 months). I do understand that median OS=17.4 months among heavily treated cohort might be better than “conventional treatment”, however, we cannot determine whether this RT+TPE was beneficial.
2. Why did authors set 1.7ng/mL as surrogate?
3. There is no description of BRAF mutation status and treatment by BRAF/MEK inhibitor.
4. There is no description of SBRT; type of radiation used or the total dose of radiation.
5. Overall, what would be the best surrogate marker for patients who may be beneficial by this strategy?
6. In supplemental figure 5, authors compared CR vs non-CR, how about responder vs non-responder?

Minor point:

1. Why did most of the patients have unspecified subtype? No record?
2. Table1: Relatlamab X>>>Relatlimab

Reviewer #2

(Remarks to the Author)

This clinical trial addresses an important clinical challenge in the treatment of metastatic melanoma today, immune checkpoint refractory disease. The authors have identified a possible way to overcome this resistance by an innovative approach combining radiotherapy with plasma exchange removing soluble PD-L1 from plasma. The obtained results in the 18 subjects enrolled are impressive with over 60% response rate and a clinical benefit rate of more than 80%.

Specific comments

P2 | 44, Abstract. The wording in this sentence should be clarified to better understand the primary objective of the trial, “Primary feasibility and efficacy endpoints of the study were adverse events (AEs) and sPD-L1 reduction by TPE.

P5 | 101. Results. It is stated that in the screened 34 patients 8 patients had a low level of sPD-L1. Are there any clinical characteristics or identified factors that differ compared to the 18 patients enrolled? Potentially there are factors of interest in the eight patients not included not related to PD-L1 where TPE still would be interesting?

When the patients were screened, were they on ICI therapy or off ICI therapy?

Why was a PD-L1 level 1.7 ng/mL or greater used as cut off for inclusion in the trial?

What organs were exposed for SBRT? It would be interesting to see if there were differences between different clinical stages M1-a-c and/or metastatic sites irradiated.

What number of TPEs were applied, three for all patients? Mean, range.

What was the duration of response until PD or death? Provide mean, range. The Figure 1F does not say if the patients were progressing and had subsequent therapy?

Did the level of sPD-L1 correlate to response not only for CR, but also between PR, SD and PD?

The study has large focus on the effects on the immune cell repertoire but how about effect on the tumour cells as a proxy for clinical efficacy? Did you eg. assess presence of ctDNA/cfDNA in the patients to correlate with the imaging data and immune cell characterization?

Table 1, Correct the spelling for Relatlimab. Change anal type to mucosal melanoma in the anal region- if that's correct?

Did the study protocol allow for a TPE also after cycle 2 of ICI?

Reviewer #3

(Remarks to the Author)

This is a well-written report of a single-arm phase II study evaluating the role of SBRT, TPE and ICI rechallenge in a patient population with unmet need, ICI-resistant melanoma. It adds to the existing literature demonstrating the importance of these soluble markers in ICI and presents a rationale strategy (ie TPE), based on prior published data, to potentially overcome ICI resistance. This study met its primary endpoint with the interventions found to be both feasible and tolerable with secondary endpoints demonstrating that OS was predicted by sPD-L1 suppression.

There have been several studies demonstrating the prognostic impact of sPD-L1 and anti tumor outcomes, in melanoma and across tumor types. The use of TPE to improve ICI outcomes is a novel concept. This study builds on the authors prior work - which demonstrated the ability to clear / reduce sPD-L1 through the use of TPE in patients with malignancy and auto-immune / inflammatory conditions.

It is interesting that while baseline sPD-L1 levels did not predict ICI response (which has been a consistent finding), the change in circulating levels did, illustrating the potential impact of TPE in reducing levels prior to ICI administration. However, one significant limitation is the inability to discern the individual effects of the three interventions performed (SBRT, TPE and ICI rechallenge) on anti-tumor outcomes.

General queries for the authors as well as by section:

- Why was the 1.7ng/mL cut-off decided? The prior data in JITC noted the 0.277 cut-off predicted OS

Patients and Results

Would be helpful to add more information on the prior ICI course (if possible) including:

- Prior best response to ICI (how many had primary vs secondary resistance)
- Number of patients progressing on ICI at time of study enrollment and median duration from last ICI treatment at time of enrollment?
- Would include in the text that the 2 pts that were treated with a new ICI were treated with single agent PD-1 (per Table 1) not dual ICI

SBRT

- Would add more data about SBRT (perhaps a supplemental table): number, site, number of remaining non-irradiated lesions
- In the text would include median number of lesions irradiated

SBRT, TPE and ICI re-challenge

- For the response rate calculations, would include in the text that irradiated lesions were excluded (as noted in methods)
- Would benefit from adding median duration of response in the text (in addition to swimmers plot)
- Any correlation with number of remaining (non-irradiated sites) with response?

Correlatives

- Would note that baseline samples were obtained after RT (per supplement figure 1); limitation that there is no pre-SBRT analysis timepoint to fully gauge benefit of SBRT on T cell / immune cell populations (if these samples existed would be beneficial to examine pre-SBRT levels)
- Did the patients with BoR / longest duration have both the T effector T cells changes AND decline in sPD-L1 and vice versa or was sPD-L1 levels alone more predictive?

Discussion

Line 233 – would specify the authors hypothesis that RT was responsible for these changes?

The primary limitation of this dataset is the inability to determine the individual contributions (+/- synergy) from the 3 interventions in OS rates. It would have been informative to demonstrate the impact of TPE and ICI-rechallenge to establish the benefit of removal of sPD-L1 on establishing (re-establishing in some cases) and ICI response and then (or in a parallel cohort) examine SBRT, TPE and ICI. Without that data the authors should expand their discussion:

1. Would also include the authors prior data in the discussion about the rate of sPD-L1 drop (also ~70% in ref 22) to highlight that TPE is felt to be the primary driver to the decline, not SBRT or ICI (though important to reference the inability to fully discern as sPD-L1 is thought to primary be derived by viable cells and SBRT and ICI have the ability to kill tumor cells and there lead to decrease in sPD-L1)
2. In addition to the SWOG data already included which supports this strategy has a higher response than salvage dual ICI, would add data supporting that the ORR of this combination (SBRT, TPE and ICI) would be higher than expected from SBRT and ICI as well as simply ICI retreatment after a period off ICI (particularly as these are patients that were progressing, not stopped due to irAEs)
3. Would also appreciate if the authors could elaborate on why the transient drop in sPD-L1 would lead to sustained responses as the majority of patients did have rebound in circulating levels (ie why post-TPE change in sPD-L1 (and TIM3) impacts OS but sPD-L1 pre-C2 does not?)
4. Elaborating on the prior point in results regarding if the populations with T cell activation / favorable immune profiles and sPD-L1 suppression were the same. Perhaps that is why only a transient drop in sPLD-L1.
5. Any clinical features that predict response / resistance to this strategy?

Grammatical suggestions

Supplement figure 1

- Would be change IO to ICI for consistency

Table 1:

- Typo: LDL at first recurrence or metastatic diagnosis; Relatlimab
- Would * at bottom of the table and include number of dual ICI pts.

Table 2

- Would consider changing hepatic failure to LFT abnormality

Reviewer #4

(Remarks to the Author)

This is an open-label single-arm phase 2 study of SBRT and TPE therapy for the immunotherapy-refractory melanoma patients. The primary endpoints are AE and sPD-L1 reduction, secondary endpoints include OS, ORR, PFS. This manuscript well described the study rationale, design and results.

1. The reported analysis is intend-to-treat (ITT), how about per-protocol(PP)? 1 subject didn't finish TPE, 16 received ICI again. Can the patients with unfinished TPE/ICI be labeled in swimmer plot 1F?
2. In the study protocol, it is said "no power calculation needed for this feasibility study". Is there post hoc power analysis to justify the conclusion that levels of sPD-L1 were significantly reduced by TPE?
3. PFS and OS are defined as time from study registration in protocol. The definitions should be described in manuscript as well. Does it vary by patients from registration to SBRT & TPE? The median OS is 17.4 months, what's the median follow-up time?
4. There are several timepoints in the study: before TPE, after TPE and before ICI1 (post-TPE), after ICI1 and before ICI2 (pre-C2). Some places also uses "baseline", "pre-treatment", "second cycle ICI re-challenge", please clearly define those timepoints and use it consistently. For example, line 132 "median recovery rate of sPD-L1 at second cycle on ICI re-challenge", is it before ICI2 or after ICI2?
5. Fig 2BCDEF, what are cutoffs for high vs low, increase or decrease? please clarify in the figure and main text. So many markers were tested, is there type I error rate control?
6. Fig 3A, volcano plot please add p-value 0.05 horizontal line. Some of the red dots are not significant at 0.05 level. Fig 3BDEGH, how high & low are classified? Please clarify.

Version 1:

Reviewer comments:

Reviewer #2

(Remarks to the Author)

Thank you for the opportunity to review the revised version of the ms.

The authors have made significant improvements of the ms following the comments raised by all the reviewers especially with adding new and relevant data in text and in supplementary tables.

I don't have any additional comments, the responses and updates are satisfying.

In the near future, I hope to see a follow up with a controlled study with the option for several TPEs and using an adequate comparator .

Reviewer #3

(Remarks to the Author)

The authors have thoughtfully and thoroughly addressed the critiques / suggestions of all the reviewers - including myself - to the best of their ability with the data available.

No further revisions suggested prior to publication.

Reviewer #4

(Remarks to the Author)

Thanks for clarification of my questions regarding the study design, endpoints and analysis. I have no further concerns.

Thank you for the opportunity to revise our manuscript “Plasma exchange and radiation resensitize immunotherapy-refractory melanoma.” We thank the reviewers for their excellent comments, encouragement, and feedback. We have made the following revisions based on their individual feedback below, listed point-by-point and marked as changes in the updated manuscript.

Reviewer #1 (Remarks to the Author): with expertise in melanoma, therapy

Authors showed that RT+TPE might restore the sensitivity of ICI in patients with ICI-resistant melanoma. More than half of patients with advanced disease treated with ICI became resistant and have few choices of further treatment. I have several questions to authors as follows:

Major point:

1. Although the ORR seemed to be high (63%), the duration of response was short (median PFS <4 months). I do understand that median OS=17.4 months among heavily treated cohort might be better than “conventional treatment”, however, we cannot determine whether this RT+TPE was beneficial.

Thank you. We agree that it is not possible in our study design to determine whether RT+TPE is truly beneficial; most responses were quite short. Another reviewer asked for a DOR analysis, which we added. We have clarified this further in the discussion section in this revised version, including these added statements:

“While the clinical activity observed in this study is favorable and warrants further study, most patients experienced disease progression and death. Although the OS we observed appears favorable in this heavily pre-treated cohort, it is uncertain whether SBRT and TPE was beneficial without a direct comparator cohort.”

and

“this trial included neither an SBRT-free control nor a TPE-free comparator arm from which to draw... conclusions. Such a study would help determine the relative contribution of each intervention, which is not possible in the present study. Our previous studies of SBRT alone did not show significant short-term changes in sPD-L1,²⁴ but direct tumor killing would be expected to reduce sPD-L1 production. In contrast, our preliminary study showed that TPE directly removes sPD-L1 from circulation.²² Thus, a potential synergy of SBRT and TPE is that TPE may eliminate circulating factors and SBRT may reduce the long-term *de novo* production of these factors from tumors. Additional study is needed.”

We look forward to additional studies with comparison arms in which more direct conclusions can be drawn.

2. Why did authors set 1.7ng/mL as surrogate?

This is an important question, thank you. Although 0.277ng/ml predicted outcomes in our previous publication, 1.7ng/mL was found to be a predictor of the poorest outcomes in patients with ICI-resistant disease in a preliminary cohort of patients with ICI-refractory melanoma. This was borne out in our study, where most of the patients we screened had much higher levels of sPD-L1 at screening. We were aware at study design that this could skew our results toward a more recalcitrant set of patients with melanoma, but that fit our proposed mechanism of action with TPE.

3. There is no description of BRAF mutation status and treatment by BRAF/MEK inhibitor.

Thank you, we went through all patients and added BRAF mutation status to Table 1 and Figure 1F as well as both

BRAF mutation status and include prior BRAF/MEK inhibitor treatments as patient-level data in Data File 1.

4. There is no description of SBRT; type of radiation used or the total dose of radiation.

Thank you, this is an important matter of clarification for the study. We have added a new table 2 to provide information about SBRT treatment and supplemental table 1 listing all radiation parameters as requested.

5. Overall, what would be the best surrogate marker for patients who may be beneficial by this strategy?

What a wonderful question!

Because the ICI commonly used in this setting directly targets the PD-1/PD-L1 axis, we feel that sPD-L1 is likely to be the best in this patient population. This gets back to your second point regarding what the appropriate cutoff of sPD-L1 would be.

Although we used sPD-L1 to select patients we thought would likely benefit (and who, based on our prior work, are likely to have the worst outcomes) we do not have a comparator arm and we found that there are *many* immunosuppressive substances in the blood of patients with melanoma. Would patients with lower sPD-L1 also benefit? Would patients with high sPD-L1 benefit with the use of anti-LAG3 ICI? We hope to answer these additional questions in future studies.

As a caveat, because the biomarkers we looked at in figure 3 were exploratory, we do *not* believe that these should be taken now as markers for likely success. Certainly, there are some immunosuppressive mediators in the blood of these patients that likely contribute to overall immunosuppression, but it would take a very large study to appropriately power answers to that question.

This also begs the question of whether a single set of TPE is enough to remove immunosuppressive mediators from the blood. Our results suggest that this is not the case. Additional studies are needed to confirm and extend these findings. We have updated the Discussion section to better clarify our thinking on this point.

6. In supplemental figure 5, authors compared CR vs non-CR, how about responder vs non-responder?

We went back to answer this question and while the median in responders was less than half of non-responders, this was not statistically significant. We have added this figure to supplemental figure 5 and to the text:

“Post-TPE sPD-L1 level comparisons for responders (i.e., CR and PR) versus non-responders (i.e., SD and PD) were not statistically significant (Supplementary Fig 5B).”

Minor point:

1. Why did most of the patients have unspecified subtype? No record?

Most of the patients in this study had metastatic disease at diagnosis, where a subtype is commonly not specified by our pathologists. This is a limitation of our study.

2. Table1: Relatlamab X>>Relatlimab

Thank you for finding this typo, we have resolved it.

Reviewer #2 (Remarks to the Author): with expertise in melanoma, therapy

This clinical trial addresses an important clinical challenge in the treatment of metastatic melanoma today, immune

checkpoint refractory disease. The authors have identified a possible way to overcome this resistance by an innovative approach combining radiotherapy with plasma exchange removing soluble PD-L1 from plasma. The obtained results in the 18 subjects enrolled are impressive with over 60% response rate and a clinical benefit rate of more than 80%.

Specific comments

1. P2 1 44, Abstract. The wording in this sentence should be clarified to better understand the primary objective of the trial, “Primary feasibility and efficacy endpoints of the study were adverse events (AEs) and sPD-L1 reduction by TPE.

Thank you, we have clarified by adding the sentence to the abstract:

“The primary objective of the study was to determine the feasibility of TPE in ICI-resistant melanoma.”

2. P5 1 101. Results. It is stated that in the screened 34 patients 8 patients had a low level of sPD-L1. Are there any clinical characteristics or identified factors that differ compared to the 18 patients enrolled? Potentially there are factors of interest in the eight patients not included not related to PD-L1 where TPE still would be interesting?

This is an excellent question! There are likely additional factors of interest in these patients not related to PD-L1 where TPE would still be interesting. However, they were excluded from the trial due to low sPD-L1 levels. Thus, we do not have longitudinal samples and did not perform prospective clinical data collection in order to further determine whether their outcomes would be improved. We believe the best way to answer these questions is in a prospective, randomized trial in which patients with low sPD-L1 are included, and we are working on that trial now.

3. When the patients were screened, were they on ICI therapy or off ICI therapy?

16/18 patients were currently on failing ICI therapy at the time of enrollment. We have added this to the manuscript under patient characteristics as below and the patient-level data will be available to readers in Data File 1:

“Sixteen patients (89%) were on current failing ICI therapy at enrollment, while two patients (11%) were on an alternate failing therapy at the time of enrollment but had at least one prior failed ICI therapy.”

4. Why was a PD-L1 level 1.7 ng/mL or greater used as cut off for inclusion in the trial?

This is an important question, thank you. Although 0.277ng/ml predicted outcomes in our previous publication, 1.7ng/mL was found to be a predictor of the poorest outcomes in patients with ICI-resistant disease in a preliminary cohort of patients. This was borne out in our study, where most of the patients we screened had much higher levels of sPD-L1 at screening.

5. What organs were exposed for SBRT? It would be interesting to see if there were differences between different clinical stages M1-a-c and/or metastatic sites irradiated.

We collected data on all sites of SBRT and added them in part to the new table 2 as well as the new supplementary table 1. We have also added them to Data File 1 for any further interest.

6. What number of TPEs were applied, three for all patients? Mean, range.

17 of 18 patients received 3 courses of TPE. One patient received 2 courses of TPE. We have clarified this in the manuscript and added it in the new Table 2 and in Figure 1F.

7. What was the duration of response until PD or death? Provide mean, range. The Figure 1F does not say if the patients were progressing and had subsequent therapy?

We have calculated mean and median (including range) duration of response. We added markers of progression to Figure 1F as requested and added the analysis as follows in the text:

“In an exploratory analysis, we measured duration of response (DOR). Mean DOR was 7.7 months (median 4.6 months, [95% CI 3.3-NR]).”

We have also highlighted this limited duration of response in the discussion.

8. Did the level of sPD-L1 correlate to response not only for CR , but also between PR, SD and PD?

This is an excellent question! We added Supplemental Figure 5B to show not only CR but also CR/PR versus SD/PD. While there was a trend toward increased levels, this was not statistically significant. We added this information to the manuscript as follows:

“Post-TPE sPD-L1 level comparisons for responders versus non-responders were not statistically significant (Supplementary Fig 5B).”

9. The study has large focus on the effects on the immune cell repertoire but how about effect on the tumour cells as a proxy for clinical efficacy ? Did you eg. assess presence of ctDNA/cfDNA in the patients to correlate with the imaging data and immune cell characterization?

Thank you, you are right that we focused on the effects on the immune cells in this study. We have *not* assessed the presence of ctDNA/cfDNA in these patients to correlate with these data in this manuscript, but we have found a collaborator and look forward to presenting these results in a future manuscript.

10. Table 1, Correct the spelling for Relatlimab.

Typo corrected

11. Change anal type to mucosal melanoma in the anal region- if that's correct?

That is correct, we have made this change.

12. Did the study protocol allow for a TPE also after cycle 2 of ICI?

Unfortunately no, this study protocol did not allow for TPE after cycle 2. This is a major limitation of this study given that a rebound of sPD-L1 correlated with poor outcomes. We clarified this in the manuscript. We are developing a follow-up trial with multiple combined TPE/ICI cycles to overcome this issue.

Reviewer #3 (Remarks to the Author): with expertise in melanoma, therapy

This is a well-written report of a single-arm phase II study evaluating the role of SBRT, TPE and ICI rechallenge in a patient population with unmet need, ICI-resistant melanoma. It adds to the existing literature demonstrating the importance of these soluble markers in ICI and presents a rationale strategy (ie TPE), based on prior published data, to potentially overcome ICI resistance. This study met its primary endpoint with the interventions found to be both feasible and tolerable with secondary endpoints demonstrating that OS was predicted by sPD-L1 suppression.

There have been several studies demonstrating the prognostic impact of sPD-L1 and anti tumor outcomes, in melanoma and across tumor types. The use of TPE to improve ICI outcomes is a novel concept. This study builds on the authors prior work - which demonstrated the ability to clear / reduce sPD-L1 through the use of TPE in patients with malignancy and auto-immune / inflammatory conditions.

It is interesting that while baseline sPD-L1 levels did not predict ICI response (which has been a consistent finding), the change in circulating levels did, illustrating the potential impact of TPE in reducing levels prior to ICI administration. However, one significant limitation is the inability to discern the individual effects of the three interventions performed (SBRT, TPE and ICI rechallenge) on anti-tumor outcomes.

General queries for the authors as well as by section:

1. Why was the 1.7ng/mL cut-off decided? The prior data in JITC noted the 0.277 cut-off predicted OS

This is an important question, thank you. Although 0.277ng/ml predicted outcomes in our previous publication, 1.7ng/mL was found to be a predictor of the poorest outcomes in patients with ICI-resistant disease in a preliminary cohort of patients. This was borne out in our study, where most of the patients we screened had much higher levels of sPD-L1 at screening.

2. Patients and Results

Would be helpful to add more information on the prior ICI course (if possible) including:

- Prior best response to ICI (how many had primary vs secondary resistance)

Thank you, this is a consistent issue brought up by multiple reviewers. We have added prior best response, primary versus secondary resistance, and specific ICI regimen to both Fig 1F and Data File 1 as requested.

3. • Number of patients progressing on ICI at time of study enrollment and median duration from last ICI treatment at time of enrollment?

This is an excellent point! 16/18 patients were progressing on ICI at the time of study enrollment, and we have added this information to the manuscript and Data File 1.

4. Would include in the text that the 2 pts that were treated with a new ICI were treated with single agent PD-1 (per Table 1) not dual ICI

Thank you, we have clarified this in the text as follows:

“All patients received ICI re-challenge, and sixteen patients (89%) received the same anti-PD-1 ICI they had received in at least one prior line of therapy while two patients (11%) received a new anti-PD-1 ICI monotherapy. Fifteen patients (83%) received nivolumab.”

We also added a new Table 2 to clarify what treatments happened *on* the study (separate from Table 1, which now only reports baseline information). Thank you for pointing out this ambiguity.

5. SBRT

- Would add more data about SBRT (perhaps a supplemental table): number, site, number of remaining non-irradiated lesions

Thank you, this is an excellent point brought up by multiple reviewers. We have added this information in the text, in the new Table 2, and the new Supplemental Table 1.

6. • In the text would include median number of lesions irradiated

Thank you, we have added this to the text and in the new Supplemental Table 1.

7. SBRT, TPE and ICI re-challenge

- For the response rate calculations, would include in the text that radiated lesions were excluded (as noted in methods)

Thank you, we have added this sentence to clarify:

“Only non-irradiated lesions were used to determine response (*i.e.*, irradiated lesions were excluded from analysis per RECIST standards).”

8. • Would benefit from adding median duration of response in the text (in addition to swimmers plot)

Thank you, we have updated the swimmer’s plot to show progression and added duration of response in the text as requested:

“In an exploratory analysis, we measured duration of response (DOR). Mean DOR was 7.7 months (median 4.6 months, [95% CI 3.3-NR]).”

9. • Any correlation with number of remaining (non-irradiated sites) with response?

This is an excellent question, and one that can be partially answered with the available data: There did not seem to be any correlation between % of sites irradiated or number of remaining unirradiated lesions and treatment response in our study. However, we have not added this analysis to the manuscript because we worry that we did not prospectively collect volume of disease for any of these timepoints, so the conclusion that number of remaining non-irradiated sites does not matter would be potentially faulty. Intuitively, it seems likely that overall burden of disease would contribute to outcomes and, while not supported by our data, cannot be excluded given this small sample size. We do report the raw data in Data File 1.

10. Correlatives

- Would note that baseline samples were obtained after RT (per supplement figure 1) ; limitation that there is no pre-SBRT analysis timepoint to fully gauge benefit of SBRT on T cell / immune cell populations (if these samples existed would be beneficial to examine pre-SBRT levels)

We apologize for this ambiguity in our manuscript. We did indeed obtain baseline samples before RT for patients going into the study, but we presented these timepoints haphazardly. We apologize, as we did not mean to mislead the reviewers. To resolve this issue, we updated Supplementary Figure 1 as well as the text to clarify which timepoints were analyzed. In brief, as we have updated these:

“Blood samples were taken at registration before SBRT (baseline), after SBRT and before TPE (pre-TPE), after TPE and before ICI re-challenge (post-TPE), and before the second cycle of ICI re-challenge (ICI2).”

However, even with this baseline sample, we do not expect significant changes between this and the pre-TPE T cell and immune populations due to the short time frame. In our previous studies, significant changes in circulating T cell and immune populations take several weeks to develop (PMID 34593526). We have referenced these studies on the effects of SBRT on anti-tumor immunity in the manuscript to help gauge the potential benefits of SBRT versus TPE. Of course, future studies will be needed to make concrete conclusions on these effects.

Thank you for this excellent question, please accept our apologies for the confusion we caused with inconsistent naming.

11. • Did the patients with BoR / longest duration have both the T effector T cells changes AND decline in sPD-L1 and vice versa or was sPD-L1 levels alone more predictive?

This is a phenomenal question, and led to the additional analyses done in Supplementary Figure 8. In brief, a combination of low sPD-L1 after TPE and increasing T_{TR} also predicted improved OS. We have added this to the discussion in order to further clarify this question:

“Why would a transient decrease in circulating sPD-L1, TIM3, or other markers lead to ICI response? We hypothesize that reduction of these soluble factors, combined with immediate PD-1 antagonism, allows activation of key anti-tumor immune cells. The reduction in sPD-L1 and increase in circulating TTR, for example, was greatest in those patients experiencing a response. A composite of both sPD-L1 reduction and TTR changes was a better predictor of OS than either variable alone.”

Thank you again for this excellent question!

12. Discussion

Line 233 – would specify the authors hypothesis that RT was responsible for these changes?

We have adjusted this line to note that SBRT may also contribute as follows:

“It is unknown whether a combination of multiple ICI modalities and SBRT or TPE could further improve responses.”

13. The primary limitation of this dataset is the inability to determine the individual contributions (+/- synergy) from the 3 interventions in OS rates. It would have been informative to demonstrate the impact of TPE and ICI-rechallenge to establish the benefit of removal of sPD-L1 on establishing (re-establishing in some cases) and ICI response and then (or in a parallel cohort) examine SBRT, TPE and ICI. Without that data the authors should expand their discussion:
1. Would also include the authors prior data in the discussion about the rate of sPD-L1 drop (also ~70% in ref 22) to highlight that TPE is felt to be the primary driver to the decline, not SBRT or ICI (though important to reference the inability to fully discern as sPD-L1 is thought to primary be derived by viable cells and SBRT and ICI have the ability to kill tumor cells and there lead to decrease in sPD-L1)

Thank you, we agree with these major limitations to the study. The question of sPD-L1 as important, and we do believe that the rate of sPD-L1 drop—as well as its being kept low—is likely the main driver of response in combination with ICI. We have highlighted this and added to the updated discussion as follows:

“...this trial included neither an SBRT-free control nor a TPE-free comparator arm from which to draw such conclusions. Such a study would help determine the relative contribution of each intervention, which is not possible in the present study. Our previous studies of SBRT alone did not show significant short-term changes in sPD-L1,²⁴ but direct tumor killing would be expected to reduce sPD-L1 production. In contrast, our preliminary study showed that TPE directly removes sPD-L1 from circulation.²² Thus, a potential synergy of SBRT and TPE is that TPE may eliminate circulating factors and SBRT may reduce the long-term *de novo* production of these factors from tumors. Additional study is needed.”

14. In addition to the SWOG data already included which supports this strategy has a higher response than salvage dual ICI, would add data supporting that the ORR of this combination (SBRT, TPE and ICI) would be higher than

expected from SBRT and ICI as well as simply ICI retreatment after a period off ICI (particularly as these are patients that were progressing, not stopped due to irAEs)

Thank you, we have not performed other clinical studies of SBRT and ICI in widely metastatic melanoma but do have significant studies in which we test SBRT and ICI in mice. We agree that this is higher than would be expected from these treatments alone. In addition, you are quite right that there are additional studies suggesting expectations from progression off of PD-1. We have added this to the discussion as follows:

“Similarly, a recent study of ICI restart after adjuvant anti-PD-1 therapy showed no responses for patients with on-therapy progression and an approximate 40% response rate after progression off of therapy.³⁰ It is unknown whether a combination of multiple ICI modalities and SBRT or TPE could further improve responses.³⁰”

15. Would also appreciate if the authors could elaborate on why the transient drop in sPD-L1 would lead to sustained responses as the majority of patients did have rebound in circulating levels (ie why post-TPE change in sPD-L1 (and TIM3) impacts OS but sPD-L1 pre-C2 does not?)

Thank you, this is such an important point! As you suggest in #4 below, we went to determine whether there is an inverse correlation between sPD-L1 and T cell activation. There is. But, as you suggested, we also went on to determine whether the two together could predict better outcomes, and it appears that they do. We have added this in supplementary figure 8. We call this out in the results section as follows:

“A combination of low sPD-L1 after TPE and increasing TTR also predicted improved OS (p=0.001, Supplementary Fig 8).”

We have further added to the discussion:

“Why would a transient decrease in circulating sPD-L1, TIM3, or other factors lead to ICI response? We hypothesize that reduction of these soluble factors, combined with immediate PD-1 antagonism, allows activation of key anti-tumor immune cells. The reduction in sPD-L1 and increase in circulating T_{TR}, for example, was greatest in those patients experiencing a response. A composite of both sPD-L1 reduction and T_{TR} changes was a better predictor of OS than either variable alone.

While the clinical activity observed in this study is favorable and warrants further study, most patients experienced disease progression and death. Even in patients with a favorable response, there was re-accumulation of immunosuppressive factors in the bloodstream and a mean DOR of 7.7 months. It is possible that a more sustained suppression of these factors (*i.e.*, performing serial TPE prior to multiple ICI cycles) would improve the robustness of ICI re-challenge response. As a further limitation, this study did not include novel ICI classes. Thus, additional clinical trials are needed to determine how best to leverage this approach in ICI resistance.”

16. Elaborating on the prior point in results regarding if the populations with T cell activation / favorable immune profiles and sPD-L1 suppression were the same. Perhaps that is why only a transient drop in sPD-L1.

Yes, exactly! As above, we added this hypothesis to the discussion. Thank you, we believe this significantly improves the manuscript.

17. Any clinical features that predict response / resistance to this strategy?

This is an excellent question. We believe that it remains to be seen what the predictors are. The additional analysis

you suggested above was a good first step, and certainly there seems to be a correlation between those who experience consistent suppression of the soluble factors and better outcomes. Beyond those observations, it seems that our best next step in answering this question is an unselected cohort of patients with resistant disease and longer-term TPE (*e.g.*, for the first several cycles of treatment). We are designing just such a study now.

18. Grammatical suggestions

Supplement figure 1

- Would be change IO to ICI for consistency

Thank you, we have made this change in addition to clearly showing the timepoints of the study in supplemental figure 1 and throughout the manuscript.

19. Table 1:

- Typo: LDL at first recurrence or metastatic diagnosis; Relatlimab

Thank you, we have resolved these typos.

20. • Would * at bottom of the table and include number of dual ICI pts.

Thank you, we have updated to add this asterisk. We have also split off a new Table 2 to differentiate between baseline features and treatment.

21. Table 2

- Would consider changing hepatic failure to LFT abnormality

Thank you, we have made this change. Note: this is now in Table 3.

Reviewer #4 (Remarks to the Author): with expertise in biostatistics, clinical trial study design

This is an open-label single-arm phase 2 study of SBRT and TPE therapy for the immunotherapy-refractory melanoma patients. The primary endpoints are AE and sPD-L1 reduction, secondary endpoints include OS, ORR, PFS. This manuscript well described the study rationale, design and results.

1. The reported analysis is intend-to-treat (ITT), how about per-protocol(PP)? 1 subject didn't finish TPE, 16 received ICI again. Can the patients with unfinished TPE/ICI be labeled in swimmer plot 1F?

That is a fantastic question! We repeated the entire analysis by PP, excluding the patient who did not finish TPE (we also labeled this patient in the swimmer plot in 1F). Note: All 18 patients received ICI re-challenge, which we have clarified in the text and figures. The per protocol analysis results were slightly better than the ITT analysis results. However, given our study design, we did not report the PP findings in the manuscript.

2. In the study protocol, it is said “no power calculation needed for this feasibility study”. Is there post hoc power analysis to justify the conclusion that levels of sPD-L1 were significantly reduced by TPE?

Yes, thank you, we have performed a post hoc analysis and found that a delta of 4.46 (mean reduction of 77.6%) with 17 patients is powered at 100% to detect this difference. Conversely, powering a study at 90% to detect this difference would require a minimum of 3 patients. We added this to the manuscript as follows:

“In a post-hoc power analysis, a delta 4.46 (77.6% reduction) would be detected at 100% power in this

study. Conversely, powering at 90% would require a minimum of three patients.”

3. PFS and OS are defined as time from study registration in protocol. The definitions should be described in manuscript as well. Does it vary by patients from registration to SBRT & TPE? The median OS is 17.4 months, what's the median follow-up time?

Because this study is focused on the effects of TPE and ICI re-challenge, PFS and OS were reported as time from TPE. Patients experienced little lag time between registration, start of SBRT, and TPE. Not surprisingly, analyses defining time to endpoints from registration found similar results. We have updated the manuscript to better reflect this methodology and include data in Data File 1.

As requested, we have added median follow-up time for both OS and PFS to the manuscript.

4. There are several timepoints in the study: before TPE, after TPE and before ICI1 (post-TPE), after ICI1 and before ICI2 (pre-C2). Some places also uses “baseline”, “pre-treatment”, “second cycle ICI re-challenge”, please clearly define those timepoints and use it consistently. For example, line 132 “median recovery rate of sPD-L1 at second cycle on ICI re-challenge”, is it before ICI2 or after ICI2?

Thank you, this is an important point of clarification. We have gone through the text to better standardize timepoint designations as follows:

“Blood samples were taken at registration (baseline) before SBRT, after SBRT and before TPE (pre-TPE), after TPE and before ICI re-challenge (post-TPE), and before the second cycle of ICI re-challenge (ICI2).”

We also updated the treatment schema in Supp Fig 1. This change will help readers easily understand the comparisons made, thank you.

5. Fig 2BCDEF, what are cutoffs for high vs low, increase or decrease? please clarify in the figure and main text. So many markers were tested, is there type I error rate control?

Cutoffs are in supplemental tables (now S4 and S5), which we have clarified in the figure and main text. We did not previously perform multiple testing correction in our initial analysis. We have now updated the supplemental tables with Hochberg adjustment on the pertinent variables. Cutoffs were performed both at balanced cutoffs (high/low) as outlined in supplemental table 4 and as increasing/decreasing in supplemental table 5. We updated the manuscript to include these clarifications and additions.

7. Fig 3A, volcano plot please add p-value 0.05 horizontal line. Some of the red dots are not significant at 0.05 level. Fig 3BDEGH, how high & low are classified? Please clarify.

We have added the p-value 0.05 horizontal line as requested. You are correct; in fact, none of the soluble factors are significantly increased in our comparison. This likely because of the small number of healthy control samples. We have clarified this in the text.